# Anti-Infective and Anti-Inflammatory Mode of Action of Peptide 19-2.5

**DOI:** 10.3390/ijms22031465

**Published:** 2021-02-02

**Authors:** Lena Heinbockel, Günther Weindl, Wilmar Correa, Julius Brandenburg, Norbert Reiling, Karl-Heinz Wiesmüller, Tobias Schürholz, Thomas Gutsmann, Guillermo Martinez de Tejada, Karl Mauss, Klaus Brandenburg

**Affiliations:** 1Brandenburg Antiinfektiva GmbH, c/o Forschungszentrum Borstel, Leibniz Lungenzentrum, Parkallee 10 b, D-23845 Borstel, Germany; lena@web.de (L.H.); k.mauss@asklepios.com (K.M.); 2Pharmakologie und Toxikologie, Pharmazeutisches Institut, Universität Bonn, Gerhard-Domagk-Str. 3, 53121 Bonn, Germany; guenther.weindl@uni-bonn.de; 3Forschungszentrum Borstel, Leibniz Lungenzentrum FG Biophysik, Parkallee 10, D-23845 Borstel, Germany; wcorrea@fz-borstel.de (W.C.); tgutsmann@fz-borstel.de (T.G.); 4Forschungszentrum Borstel, Leibniz-Lungenzentrum, FG Molekulare Grenzflächenbiologie, Parkallee 22, D-23845 Borstel, Germany; jbrandenburg@fz-borstel.de (J.B.); nreiling@fz-borstel.de (N.R.); 5EMC MIcrocollections, Sindelfinger Straße 3, D-72070 Tübingen, Germany; wiesmueller@microcollections.de; 6Klinik und Poliklinik für Anästhesiologie und Intensivtherapie, Universitäsmedizin Rostock, Schillngallee 35, 18057 Rostock, Germany; Tobias.Schuerholz@med.uni-rostock.de; 7Departamento de Microbiologia, Universidad de Navarra, Irunlarrea 1, E-31008 Pamplona, Spain; gmartinez@unav.es; 8Asklepios-Klinik Hamburg-Altona, Paul-Ehrlich-Str. 1, D-22763 Hamburg, Germany

**Keywords:** Aspidasept, lipopolysaccharide, antimicrobial peptides, pathogen-associated patterns, pharmacodynamics, inflammation, sepsis

## Abstract

The polypeptide Pep19-2.5 (Aspidasept^®^) has been described to act efficiently against infection-inducing bacteria by binding and neutralizing their most potent toxins, i.e., lipopolysaccharides (LPS) and lipoproteins/peptides (LP), independent of the resistance status of the bacteria. The mode of action was described to consist of a primary Coulomb/polar interaction of the N-terminal region of Pep19-2.5 with the polar region of the toxins followed by a hydrophobic interaction of the C-terminal region of the peptide with the apolar moiety of the toxins. However, clinical development of Aspidasept as an anti-sepsis drug requires an in-depth characterization of the interaction of the peptide with the constituents of the human immune system and with other therapeutically relevant compounds such as antibiotics and non-steroidal anti-inflammatory drugs (NSAIDs). In this contribution, relevant details of primary and secondary pharmacodynamics, off-site targets, and immunogenicity are presented, proving that Pep19-2.5 may be readily applied therapeutically against the deleterious effects of a severe bacterial infection.

## 1. Introduction

In previous investigations, a wide variety of anti-infective and anti-inflammatory activities of the peptide Pep19-2.5 against bacterial pathogens were reported [1]. Among these data, the neutralization of the bacterial toxins is of particular importance, since these molecules are the triggering factors of severe inflammatory reactions (e.g., sepsis) in mammals [2]. Recognition of pathogens by the host is mediated by the so-called Pattern Recognition Receptors (PRR), which are expressed on the surface of immune cells. Specific binding of PRR to pathogen-associated molecular patterns (PAMPs) expressed on the microbial surface leads to the activation of the inflammatory response and the secretion of pro-inflammatory cytokines such as IL-1, IL-6, and TNF-alpha. The peptide Pep19-2.5, belonging to the group of synthetic anti-lipopolysaccharide peptides (SALP), not only blocked the action of the most potent PAMPs from Gram-negative bacteria (i.e., lipopolysaccharides (LPS)), but also those expressed by their Gram-positive counterparts (lipoproteins/peptides (LP)) [3,4]. Interestingly, it was shown that Pep19-2.5 does not block only the pro-inflammatory activity of LPS in its native location (i.e., as part of the outer membrane of whole bacterial cells) but also in free form (i.e., released in the extracellular medium) [5]. Even more importantly, *Aspidasept* was reported to efficiently neutralize the pro-inflammatory activity of LPS-carrying outer membrane vesicles (OMVs) [6]. Precisely, the shedding of OMVs by Gram-negative bacteria has been identified as the natural mechanism of LPS released from the bacterial surface [4].

In addition to PAMPs, also the own host’s danger- or damage-associated molecular patterns (DAMPs) can play a role as has been proven for inflammation in murine cardiomyocytes stimulated by human sepsis serum [7]. Here, heparan sulfate as a DAMP was shown to be responsible for the inflammation and could be blocked by the addition of Pep19-2.5.

In previous papers, possible mechanisms of interaction between Pep19-2.5 and pro-inflammatory toxins were discussed, but a complete picture of the mode(s) of action(s) is still lacking. Moreover, in previous reports, we used as our model toxin the LPS from *Salmonella minnesota* rough mutant R60, a strain corresponding to the Ra-type, an LPS being described as the bioactive moiety within wild-type strains (S-LPS) [8]. Here, we extend the mechanisms of action of toxin-neutralization to other LPS from mutant strains as well as to non-LPS stimulants found in Gram-positive strains and mycoplasmas. Furthermore, the anti-endotoxic effect of Pep19-2.5 in the presence of human mononuclear cells (hMNCs) is analyzed under very diverse conditions including (i) a delayed application of the peptide (“therapeutical approach”), (ii) removal of the peptide from the serum after incubation, (iii) after incubation in plasma, and (iii) in the presence of antibiotics. Similarly, the influence of components of the coagulation system (heparin and anti-thrombin-III) is analyzed.

## 2. Results

### 2.1. Neutralization of LPS with Varying Length of the Sugar Chain

A previous investigation has shown that LPS from wild-type strains such as LPS O55:B5 from *Escherichia coli* and LPS from *Salmonella abortus equi* as well as those from a Ra-mutant R60 from *Salmonella minnesota* can be effectively neutralized by the addition of Pep19-2.5 [9].

The latter LPS was taken in particular because it represents the bioactive substructure within heterogeneous wild-type forms. Therefore, it is of interest whether LPS rough mutant strains with significantly shorter sugar parts can also be neutralized effectively. LPS were chosen that have the lowest possible sugar content (deep rough mutant LPS) Besides the lipid A part, there are only two (LPS from WBB01) or 2 to 3 (LPS from R595) Kdo (2-keto-3-deoxyoctonate) units bound to lipid A present. In Figure 1, the results are presented from two deep rough mutants, R595 from *Salmonella minnesota* and WBB01 from *Escherichia coli*, in comparison to the results for LPS from *S. minnesota* R60. It can be seen that Pep19-2.5 is similarly active for all LPS, although the absolute values of the cytokine are considerably higher for LPS with the longer sugar chain.

The former two LPS consist essentially of lipid A and two 2-keo-3-deoxyoctonate (Kdo) molecule units; the latter LPS has a complete core oligosaccharide lacking only the O-antigen characteristic for smooth-form (wild-type) LPS.

In a similar way, Pep19-5 was shown to be able also to neutralize wild-type LPS, as published for those from *Escherichia* O55:B5, *Salmonella minnesota* and *Salmonella abortus equi*, and *Citrobacter freundii* [9].

### 2.2. Immunization Experiments

An important point is the question of whether peptides are immunogenic when administered to mammals. To test this, experiments were performed by Charles River using the KLH (Keyhole Limpet Hemocyanin)-peptide19-2.5 carrier conjunction and immunization of rabbits.

Results of the analysis of the binding of rabbit serum drawn at different timepoints (before immunization and at days 38 and 70) are presented in Figure 2. The isothermal titration plots are indicative of a relatively unspecific interaction, suggesting that the peptide has low immunogenic potential.

### 2.3. Interaction with Antibiotics: LPS

As a candidate for anti-sepsis therapy, Pep19-2.5 would be, in all likelihood, administered in combination with antibiotics (AB). The success of such an approach has already been shown in mouse and rabbit models of sepsis [1,10]. These experiments were performed with different mixtures of LPS, Pep19-2.5, and the antibiotics tetracycline, ciprofloxacin, ceftriaxone, and amoxicillin at different concentration ratios.

For the antibiotic ceftriaxone (Cef) and Cef: Pep19-2.5 mixtures, SAXS data were also generated in the presence of LPS to find out how the AB may influence its aggregate structure. For LPS alone, a non-lamellar bi-layered structure was found, as has been reported before [11] (not shown). This aggregate structure does not change as indicated in Figure 3, showing only a broad intensity distribution between 0.1 and 0.3 nm^−1^ characteristic for an LPS bilayer, not allowing a more detailed characterization. In the presence of the peptide, the patterns change as evidenced by the occurrence of two sharp reflections at 9.90 and 4.50 nm corresponding to the first and second order of a multilamellar organization of the LPS, as found earlier [12]. This is not influenced by the presence of ceftriaxone (Figure 3), showing that the AB does not interfere with the structural changes of LPS induced by Pep19-2.5.

The same experiments of the LPS-induced cytokine production of pure LPS, pure antibiotic, and 1:1 mixtures of Pep19-2.5 and AB show clearly that the antibiotics alone are not able to reduce the LPS-induced cytokine production (Figure 4A). On the contrary, there is even a slight increase in cytokine production. In the presence of Pep19-2.5, there is a clear inhibition of the secretion of TNFα, in particular for the LPS concentration of 1 ng/mL. For the Pep19-2.5:AB mixtures, it is observed that inhibition of the cytokine secretion is nearly as high as found for the action of the peptide alone (Figure 4C). Similar results were found for the experiments performed in the presence of ciprofloxacin, tetracycline, and amoxicillin. These results demonstrate that no influence of the antibiotics on the inhibiting action of the peptide takes place.

### 2.4. Interaction with Antibiotics: Bacteria

In the next step, the effect of a combined interaction of Pep19-2.5 with antibiotics from two different classes, ceftriaxone, and ciprofloxacin to inhibit the cytokine induction in hMNC, was investigated with complete bacteria (10^6^/ml *Salmonella minnesota* strain R60). The data (Figure 5) show that the antibiotics alone only marginally influence the cytokine release from the hMNCs. The addition of Pep19-2.5, however, significantly decreases the TNFα concentration dose-dependently. This means that for both antibiotics, the presence of the peptide leads to an inhibition of the inflammation-inducing activity of the bacteria, most probably due to the LPS-blocking activity.

### 2.5. Cell Stimulation after Removal of Free Pep19-2.5

Previous investigations have shown that LPS may stimulate immune cells after intercalation into their cytoplasmic membrane [1]. To test whether Pep19-2.5 may also act at the membrane level, cells were incubated with Pep19-2.5 for 30 min, and then, cells were washed for 30 min to remove any peptide. After that, cells were stimulated with LPS. The data in Figure 6 show that there is still an essential inhibition of the LPS-induced cytokine secretion despite the absence of free Pep19-2.5. This means that (i) an essential step in cell stimulation by LPS takes place at the membrane level and (ii) peptide intercalates into the membrane and inhibits LPS to bind to the cell receptors such as TLR4/MD2.

### 2.6. Cell Stimulation after Incubation in Human Blood Plasma

To test a potential inactivation of the peptide in human blood, Pep19-2.5 was incubated at 37 °C for different times, i.e., 5, 10, 15, 30, 60, and 120 min in 20% human plasma, after which stimulation with an LPS:Pep19-2.5 ratio of 1:10 weight % was performed. Furthermore, the peptide in L-configuration as well as that in D-configuration, which should be resistant against the action of proteases, was compared (Figure 7).

The data in Figure 7 unequivocally show that the peptide’s ability to neutralize LPS-inducing potential is inversely proportional to the duration of its incubation in blood. However, at least for up to 1 h, there is still a strong inhibition of the LPS-induced hMNC stimulation, showing that the action of proteases or other interaction with plasma compounds does not abrogate the peptide’s activity in the short-term. This is impressively confirmed by the results for the peptide in D-configuration (Figure 7B).

### 2.7. Cell Stimulation after Delayed Addition of Pep19-2.5

First, LPS was added to 5 ×·10^6^ MNC, and then, the peptide was added at the indicated time points later. In all cases, total stimulation was 4 h, before the cytokine determination was performed. 

These assays sought to investigate whether Pep 19-2.5 could neutralize LPS-induced secretion of TNFα by human mononuclear cells when the peptide was added at different time points after LPS induction. LPS was tested at two different concentrations and the data obtained at (LPS):(Pep19-2.5) = 1:100 weight % showed that indeed, peptide was able to abrogate cytokine induction even when added 2.5h after LPS induction (Figure 8). This can be understood by assuming that the cell activation at the receptor system TLR4/MD-2 is not an irreversible process on this timescale.

### 2.8. Influence of Serum Albumin

Isothermal titration experiments were performed with 200 μg/mL HSA (human serum albumin) added to 2 mM Pep19-2.5. Interaction of the peptide with HSA was insignificant since only some slightly exothermic and noisy reactions were detected. Similarly, in the cytokine assay, the influence of HSA on the ability of Pep19-2.5 to inhibit LPS-induced secretion by MNC was negligible.

### 2.9. Interaction with Coagulation Compounds

The interaction of Pep19-2.5 with LPS in the human MNC activation test was also tested in the presence of heparin and antithrombin III, which are relevant compounds of the coagulation system and play a role in particular in the case of septic patients.

#### 2.9.1. Influence of Heparin

Heparins as endogenous sugar polymers acting as anti-coagulants are polysulfated and could, therefore, compete with the binding of Pep19-2.5 to LPS. It is worth noting that septic patients are treated daily with heparins. After stimulating hMNCs with LPS, heparin was added at different concentrations to LPS, and then, Pep19-2.5 was added. The data in Figure 9A show that, even in the presence of heparin, Pep19-2.5 retains a significant inhibitory activity, in particular at 10 and 1 ng/mL. Importantly, the daily heparin dose for septic patients (2 E.U./mL blood) is much lower than the concentrations applied here (50 E.U./mL blood).

#### 2.9.2. Influence of Anti-Thrombin (AT)-III

In a similar way, the influence of compound AT-III, an important inhibitor of the coagulation system, was analyzed (Figure 9B) and induced no weakening of the inhibiting action of the peptide.

### 2.10. Interaction with NSAID

The administration of non-steroidal anti-inflammatory drugs (NSAIDs) such as diclofenac and ibuprofen in frequent cases cannot be neglected in the case of infected patients. Therefore, the combined action of Pep19-2.5 and NSAID systems in vitro as well as in vivo (a mouse model of lethal endotoxemia) in the case of ibuprofen, a synergistic action, exhibiting a decrease in inflammation, was connected with a survival benefit of the mice (Figure 10).

These in vivo assays showed that survival was associated with a decrease in TNFα levels and also in the production of the NSAID-specific target prostaglandin-E2. A less significant effect was found for diclofenac. Therefore, the combined administration of Pep19-2.5 and NSAIDs leads to synergistic rather than antagonistic actions.

### 2.11. Off-Site Targets

The potential interaction of Pep19-2.5 on the TNFα system was exemplarily done using TNFα as a stimulant, which stimulates the cells via the TNFα-receptor and interleukin-8 levels were measured as the cellular response. The data in Figure 11 indicate that there is no effect of Pep19-2.5 on the TNFα-induced secretion of IL-8 seen.

The data indicate that the peptide does not seem to interact with TNFα, as no effect of Pep19-2.5 on the TNFα-induced secretion of IL-8 was detected.

### 2.12. Supramolecular Aggregate Structure of Non-LPS Toxins (Lipopeptides)

It has been shown that the supramolecular aggregate structure of LPS is a determinant of biological activity. A non-lamellar bi-layered structure was found to correspond to a bioactive aggregate structure, whereas a multilamellar aggregate indicates an inactive conformation [14]. To find out whether this “endotoxin supramolecular conformation” is a general principle also valid for non-LPS lipopeptide/lipoprotein (LP) structures, which are the main endotoxin structures of Gram-positive strains [3], small-angle X-ray scattering (SAXS) measurements were performed with the fibroblast-stimulating lipopeptide (FSL-1) in the absence and presence of Pep19-2.5 (Figure 12). The pure FSL-1 alone has a scattering pattern that does not fit to the reflections occurring at 1, ½, 1/3, etc., of a periodicity characteristic for a multilamellar structure. The apparent non-lamellar structure, however, cannot be assigned to a particular, for example, cubic, aggregate structure. In contrast, in the presence of the peptide, clear assignment to a multilamellar organization is possible due to the occurrence of reflections at 1 and ½ of a periodicity located at 9.45 nm.

Although the original approach of Aspidasept was directed against Gram-negative LPS, we have found that Gram-positive or mycoplasmic lipopeptides/proteins can also be neutralized in a similar way [3]. As an example, data for the inactivation of the well-known LP MALP-2 (macrophage-activating lipopeptides) and FSL-1 are shown (Figure 13), with a concentration-dependent decrease in TNFα secretion with increasing Pep19-2.5 concentrations.

### 2.13. Inhibition of Stimulation Induced by Bacterial Lipopeptides

Furthermore, in the case of cell stimulation with non-LPS lipopeptide compounds, the shortened synthetic variant Pam_3_CSK_4_ is frequently taken as a synthetic model. It must be emphasized that this LP is not a constituent occurring in nature. We have found [14] that Pep19-2.5 suppresses, if at all, only marginal Pam_3_CSK_4i_-induced cell stimulation (Figure 14). This was found to also hold true for other SALP variants. 

### 2.14. Binding Affinity to Pam_3_CSK_4_

For an understanding of this behavior, we have performed ITC measurements of the interaction of Pam_3_CSK_4_ with Pep19-2.5. This technique possesses an exquisite sensitivity to detect possible binding reactions [15]. 

The data in Figure 15 show, in accordance with the cytokine test, that no binding at all can be observed. As an explanation of the lacking interaction of Pam_3_CSK_4_ with Pep19-2.5, it should be considered that this LP has a very short hydrophilic moiety and four positively charged lysine (K) residues which do not allow binding to the polycationic N-terminal side of Pep19-2.5, which is normally the first step in the interaction of Pep19-2.5 with LPS or with other LP. 

### 2.15. Inhibition of Stimulation Induced by Mycobacterial Lipopeptide 

Furthermore, experiments with complete mycobacteria such as *Mycobacterium. tuberculosis* as well as with part structures of these were also performed. These investigations are beyond the scope of this presentation and will be published elsewhere. Here, only one example of the interaction of Pep19-2.5 with a mycobacterial lipopeptide is presented regarding the neutralization of the inflammation-inducing bLP002, a shortened variant of the 19 kDa antigen of *M. tuberculosis*. In Figure 16, an ITC experiment is shown, which illustrates the inactivation of this LP (chemical structure Pam3Cys-SSNKSTTGSGETTTA)—by adding Pep19-2.5.

## 3. Discussion

In the present study, the influence of various parameters on the action of the anti-infective and anti-inflammatory polypeptide Pep19-2.5 (Aspidasept) was analyzed. These analyses comprise investigations of primary and secondary pharmacodynamics, off-site targets, and include Gram-negative LPS (endotoxins) as well as non-LPS lipopeptides from Gram-positive or mycoplasma origin.

The data unequivocally show that Pep19-2.5 can block cytokine production induced by all kinds of pro-inflammatory toxins (Figure 1). This conclusion has reinforced the insights of previously published results on peptide interaction with wild-type LPS forms [9] as well as with bacterial lipopeptides (Figure 14, Figure 15 and Figure 16). Our data are consistent with the bioactive and non-bioactive LPS forms, consisting of a non-lamellar supramolecular aggregate [16,17] and of multilamellar aggregates [18], respectively. The results obtained with lipopeptides (Figure 13) lead to similar results. The addition of Pep19-2.5 converts the non-lamellar aggregate structure of the LP into a multilamellar one. Thus, there is a general validity of the principle that the conformation of LPS, as well as non-LPS structures from lipoproteins/lipopeptides, is in direct correlation with their corresponding endotoxicity.

It seems that the principles of inflammation induction by bacterial compounds with an amphiphilic character, which may lead to the life-threatening sepsis syndrome, and their inhibition exerted by Aspidasept, obey the same rules, although the chemical structures of the participating toxins are quite diverse. Of similar importance are these data concerning the fact that they give clear indications that aggregates of these compounds are the decisive units for bioactivity rather than monomeric units [19]. Further data discussed here confirm this assumption: As indicated in Figure 6, the peptide incorporates into target cell membranes, because they are still active in neutralizing the LPS-induced stimulation after removal of external free peptide. Such incorporation of the peptide into membranes from phosphatidylcholine or phosphatidylserine was observed earlier in experiments with Förster resonance energy transfer spectroscopy (FRET) [20].

These data can be understood by the blockade of the TLR4/MD-2 system on the level of membrane interaction and not from the outside, as has been proposed in other models with monomeric units as active units [21]. In the latter paper, Park et al. proposed with crystallographic data of the TLR-4 system that only one LPS monomer leads to cell activation. However, these are severely hampered by the fact that TLR4 without its membrane part, i.e., only the ectodomain, could be crystallized leading to unphysiological conditions. Moreover, it was reported in an investigation directly comparing aggregates and monomers in identical concentrations that only the former exhibited biological activity in the TNFα assay as well as in the *Limulus* test. [19]. Additionally, the data of the delayed action of Pep19-2.5 (Figure 8) support this interpretation, since interaction with LPS in an intramembranous manner according to a previous model [22] would make the data intelligible. The details of the cell activation mechanisms via theTLR4/MD-2 receptor (LPS) or the TLR2 (or TLR2/TLR6) receptor systems, however, are beyond the topic of this study, and will be published elsewhere.

The interaction studies, whether with different kinds of antibiotics or with NSAIDs, are indicative that in each case, Pep19-2.5 interacts in a synergistic rather than in an antagonistic mode (Figure 4, Figure 5, and Figure 11). In particular, only the peptide, and not the antibiotics, can reduce the inflammation reaction, which is responsible for the emergence of sepsis. Furthermore, the presence of antibiotics does not influence the action of Pep19-2.5, as also seen in the experiment of the aggregate structure of LPS (Figure 3), indicating a bio-inactive multilamellar structure independent of the presence of the antibiotics.

Furthermore, the presence of relatively high concentrations of compounds of the coagulation system (Figure 9 and Figure 10) only slightly reduces the activity of the peptide. Additionally, a possibly problematic influence of the peptide on the immune system such as on the TNFα receptor (Figure 12) does not take place. This is an important point regarding the frequently discussed possibility that AMP may lead to a modulation of the immune system [23]. Here, we do not see a direct influence of Pep19-2.5 on TNF-induced signaling, although we cannot rule out that the peptide may interfere with other as yet unidentified immunoregulatory pathways or proteins. So far, the mode of action of immunomodulatory actions of AMP are neither shown convincingly nor understood.

In this paper, detailed toxicologic actions of Pep19-2.5, which were shown to start in the range 30 to 50 μg/mL, are not discussed. The details of these damaging activities will be published elsewhere.

## 4. Materials and Methods

### 4.1. Materials

As LPS, those from *Escherichia coli* and *Salmonella minnesota* deep rough mutants WBB01 and R595, respectively, and from Ra-mutant R60 from *S. minnesota* were used (own extraction). Synthetic lipopeptides MALP-2 (macrophage-activating lipopeptide-2), FSL-1 (fibroblast-stimulating lipid-1), Pam_3_CSK_4_, and bLP002 shortened variant from *Mycobacterium tuberculosis* 15 kDa lipoprotein were a kind gift from EMC Microcollections (Tübingen, Germany). Heparin, anti-thrombin III, Ibuprofen, Ceftriaxone, and Ciprofloxacin were from Sigma-Aldrich Chemicals (Merck, Darmstadt, Germany).

Peptide Pep19-2.5 (Aspidasept) was synthesized as described earlier [24]. The amino acid sequence is GCKKYRRFRWKFKGKFWFWG. The sample used here (batch 1053821) was purchased from BACHEM (Bubendorf, Switzerland). The purity of the peptide could be estimated to be better than 95% by HPLC.

### 4.2. Methods

#### 4.2.1. Endotoxemia Mouse Model

The details of the endotoxemia mouse model are found elsewhere [1].

#### 4.2.2. Aggregate Structures by Small-Angle X-ray Scattering (SAXS) with Synchrotron Radiation

The X-ray scattering measurements were performed on the P12 beamline of the European Molecular Biology Laboratory (EMBL) outstation at HASYLAB on the storage ring PETRA of the Deutsches Elektronen Synchrotron (DESY) at Hamburg [9]. Briefly, scattering patterns in the range of scattering vector 0.05 < s < 1 nm^−1^ (s = 2 sin ɵ/λ, 2 ɵ is the scattering angle and λ the wavelength = 0.15 nm) were recorded with exposure times of 1min using an image plate detector with online readout (Mar345; Marresearch, Norderstedt, Germany). Further details can be found as described previously [24,25]. In the scattering patterns presented here, the logarithm of the diffracted intensities I(s) is plotted versus s. The X-ray spectra were evaluated using standard procedures, which allow the spacing ratios of the scattering maxima to be assigned to defined three-dimensional structures of the endotoxin samples in the different formulations.

#### 4.2.3. Stimulation of Human Mononuclear Cells

Mononuclear cells (MNC) were isolated from heparinized blood samples obtained from healthy donors as described previously [1,24]. The cells were resuspended in medium (RPMI 1640), and their number was equilibrated at 5 × 10^6^ cells/mL. For stimulation, 200 μL MNC (1 × 10^6^ cells) in RPMI medium supplemented with PS/Glu2% (penicillin/streptomycin/glutamine, Biochrome, Berlin, Germany) plus AB serum (4%) was transferred into the wells of a 96-well culture plate.

The formulations to be tested for stimulation were preincubated for 30 min at 37 °C and added to the cultures at 20 μL per well. The cultures were incubated for 4 h at 37 °C with 5% CO_2_. Supernatants were collected after centrifugation of the culture plates for 10 min at 400× *g* and stored at 20 °C until immunological determination of tumor necrosis factor alpha (TNF-α), carried out with a sandwich enzyme-linked immunosorbent assay (ELISA) using a monoclonal antibody against TNF (BD Biosciences, Heidelberg, Germany) as described previously in detail [1]. In the case of the stimulation by the TNFα receptor, interleukin-8 in the supernatants was determined immunologically.

#### 4.2.4. Binding Affinity Measurements by Isothermal Titration Calorimetry (ITC)

The binding of various compounds to peptide Pep19-2.5 in different formulations was analyzed by microcalorimetric measurements in the ITC200 (GE Healthcare, München, Germany), as recently described [15,24]. For this, 1 mM (2.71 mg/mL) Pep19-2.5 in buffer was titrated to the compounds to be tested at the given concentrations. The measured enthalpy changes (∆H) were recorded versus time and the peptide:compound concentration ratio.

## 5. Conclusions

In this contribution, a novel anti-infective and anti-inflammatory polypeptide Pep19-2.5 (Aspidasept) was characterized in detail, in particular concerning the pharmacodynamic behavior and the interaction with a variety of pathogen-associated molecular patterns. The data indicate the suitability of this polypeptide drug to (i) generally neutralize the PAMP-induced inflammation independently of their origin, (ii) confirm and extend the concept of the endotoxin conformation regarding the importance of the aggregate structures, (iii) exhibit convincing pharmacodynamic behavior regarding the presence of antibiotics, bacteria, or NSAIDs, and (iv) not interfere with off-site targets.

## 6. Patents

The peptides are protected by IPR: “Novel antimicrobial peptides” (WO2009/124721; priority: 04/2008); granted in EP (validated in CH, DE, ES, FR, GB), US and JP, and “Means and methods for treating bacterial infections” (PCT/EP2017/053487; priority: 02/2016). Patent pending.

## Figures and Tables

**Figure 1 ijms-22-01465-f001:**
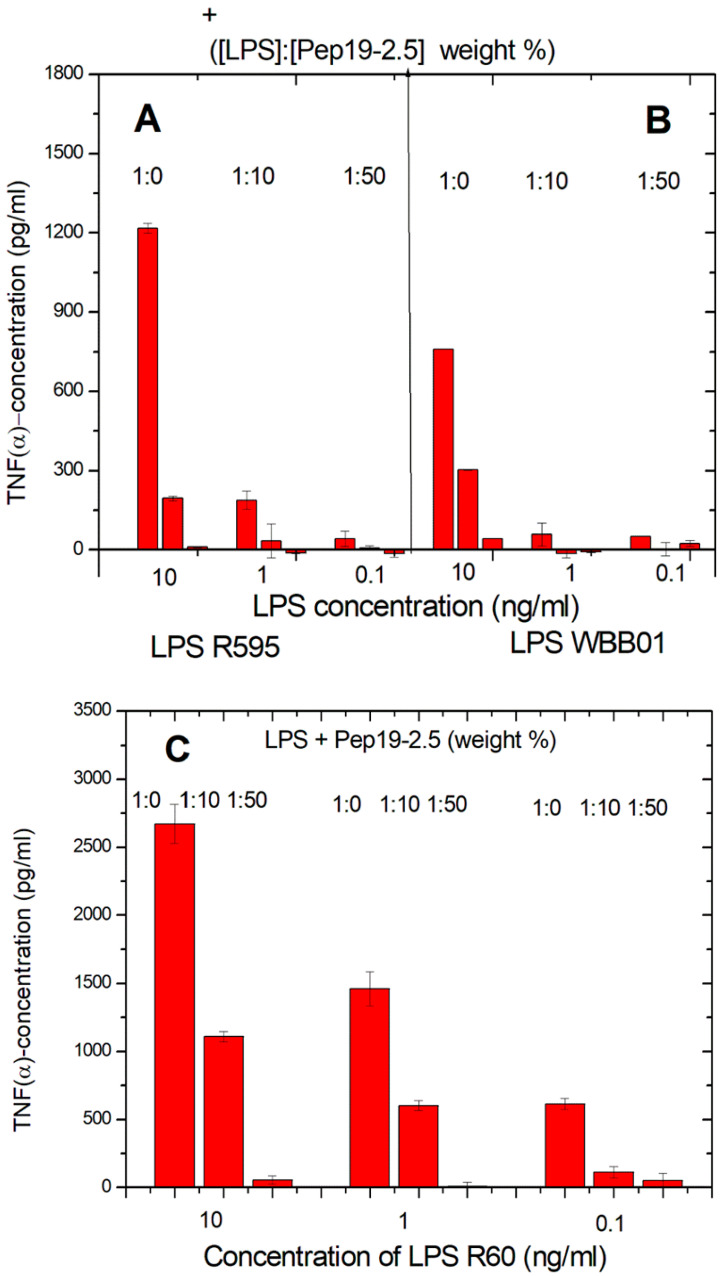
LPS-induced cytokine secretion of human mononuclear cells in the absence and presence of two concentrations of Pep19-2.5 for deep rough mutant LPS R595 (**A**), deep rough mutant LPS WBB01 (**B**), and rough mutant Ra (**C**). Mean values and standard deviations due to the determination of TNFα in duplicate.

**Figure 2 ijms-22-01465-f002:**
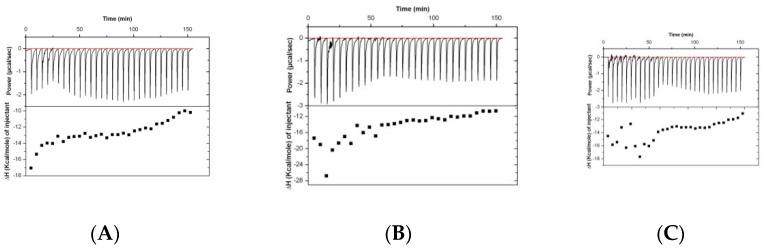
Isothermal titration (ITC) experiments performed by titrating the possibly immunized rabbit serum with Pep19-2.5. The data for three different batches (**A**–**C**) are characteristic only for a relatively unspecific interaction.

**Figure 3 ijms-22-01465-f003:**
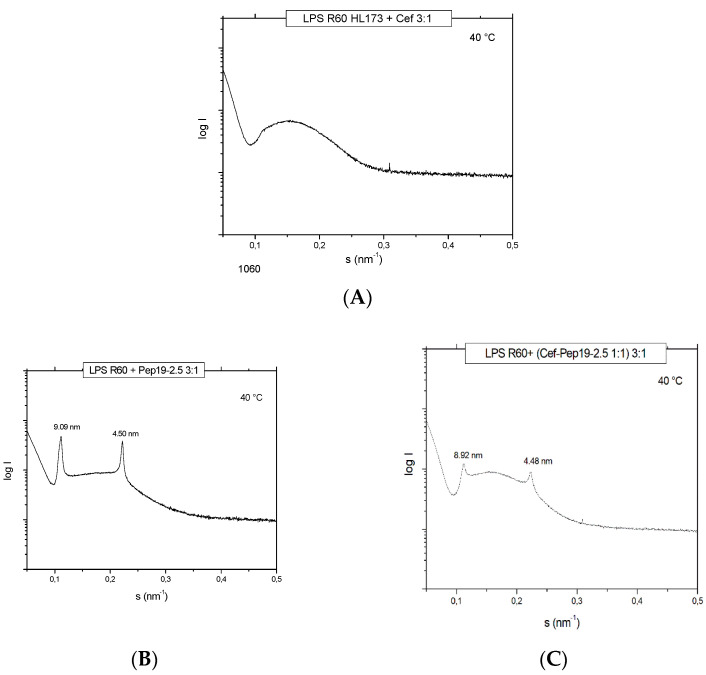
Small-angle X-ray scattering patterns at 40 °C of LPS R60 + ceftriaxone (**A**), LPS + Pep19-2.5 (**B**), and LPS + (Cef+Pep19-2.5 1:1 weight %) (**C**). Presented is the logarithm of the scattering intensity log I versus the scattering vector s (= 1/d, d spacings of the reflections).

**Figure 4 ijms-22-01465-f004:**
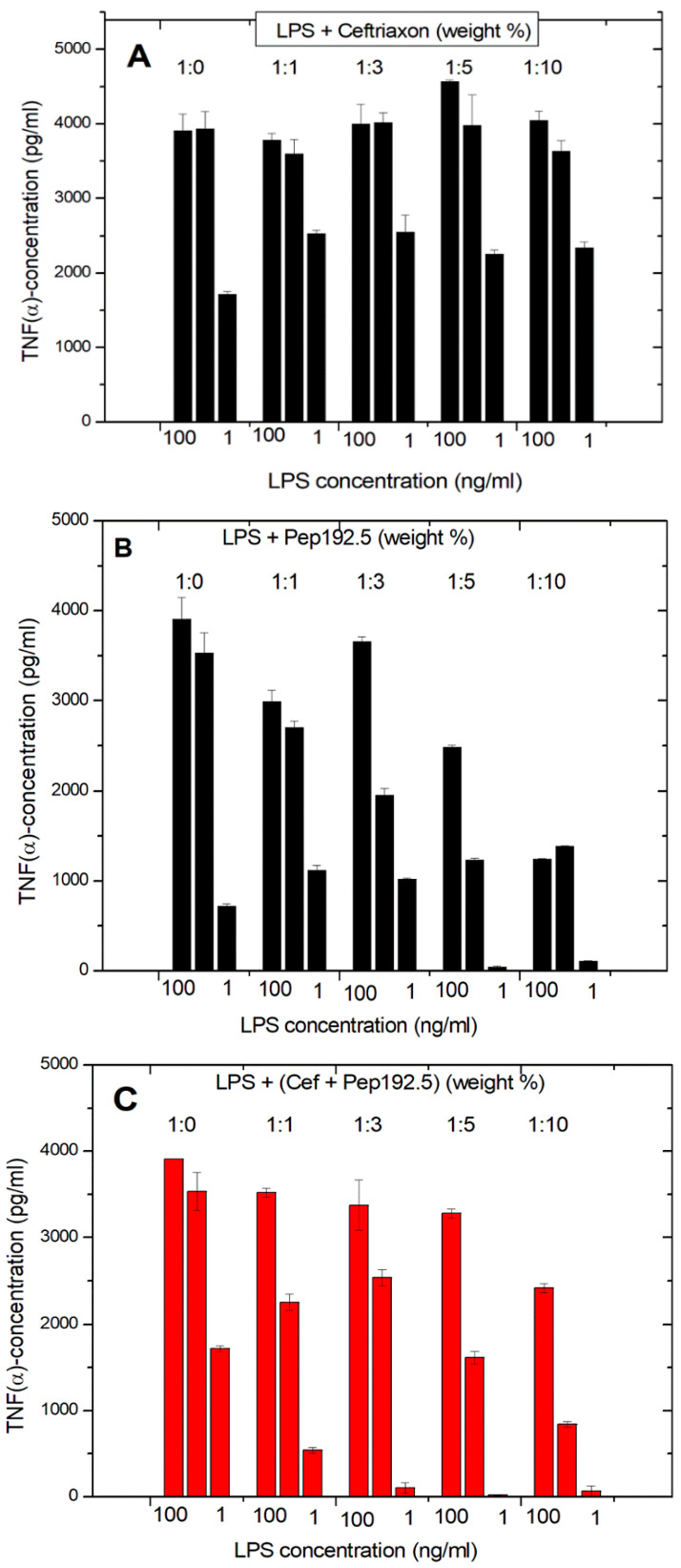
LPS-induced cytokine induction in human mononuclear cells in the presence of the antibiotic Ceftriaxone (**A**), Pep19-2.5 (**B**), and a 1:1 weight% combination of Ceftriaxone and Pep19-2.5 (**C**). Mean values and standard deviations due to the determination of TNFα in duplicate.

**Figure 5 ijms-22-01465-f005:**
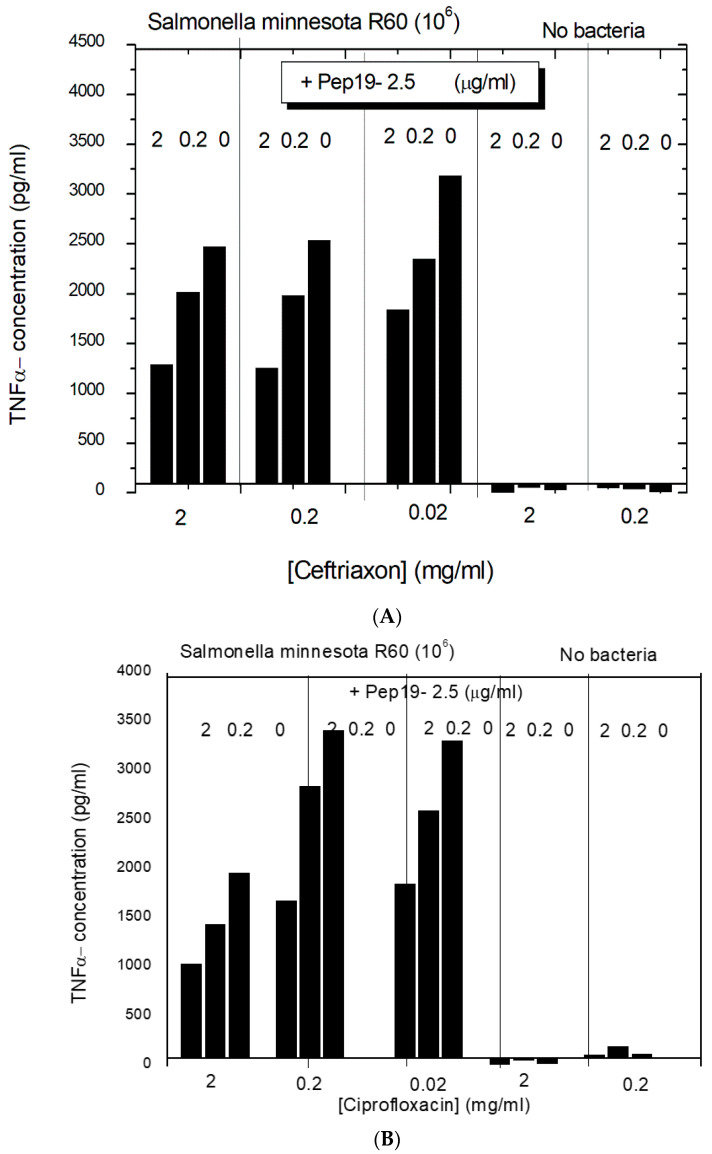
Secretion of the cytokine tumor necrosis factor α (TNFα) by human mononuclear cells exposed to bacteria from *Salmonella minnesota* strain R60 and in the presence of ceftriaxone (**A**) and ciprofloxacin (**B**) and Pep19-2.5. Mean values and standard deviations due to the determination of TNFα in duplicate.

**Figure 6 ijms-22-01465-f006:**
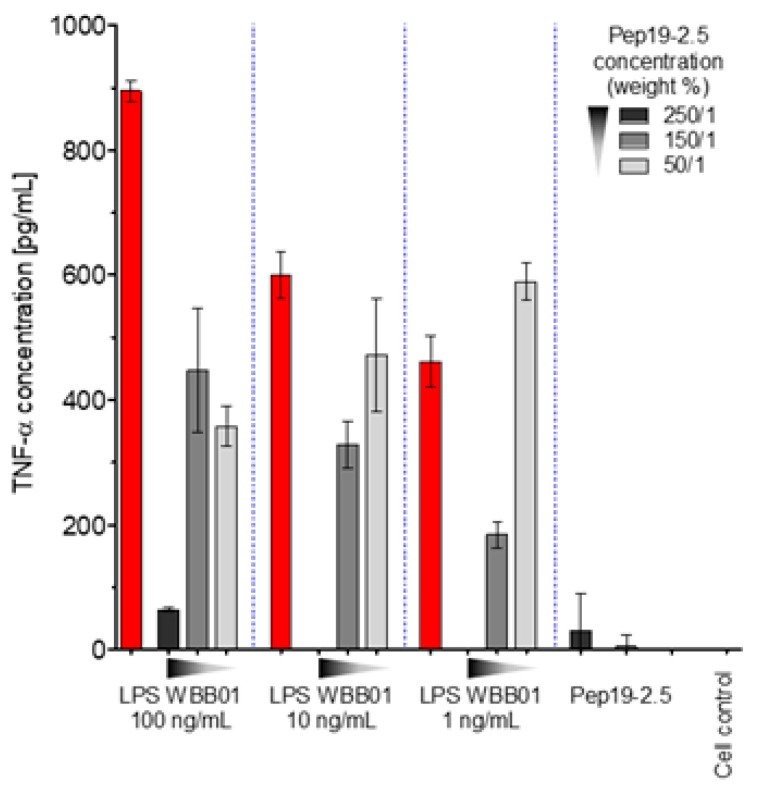
LPS-induced activation of human mononuclear cells after removal of the external peptide. Pep19-2.5 was added to the cells for 30 min, and after that, the cells were washed for 30 min to remove any free peptide. Finally, the cells were stimulated with LPS. Mean values and standard deviations due to the determination of TNFα in duplicate.

**Figure 7 ijms-22-01465-f007:**
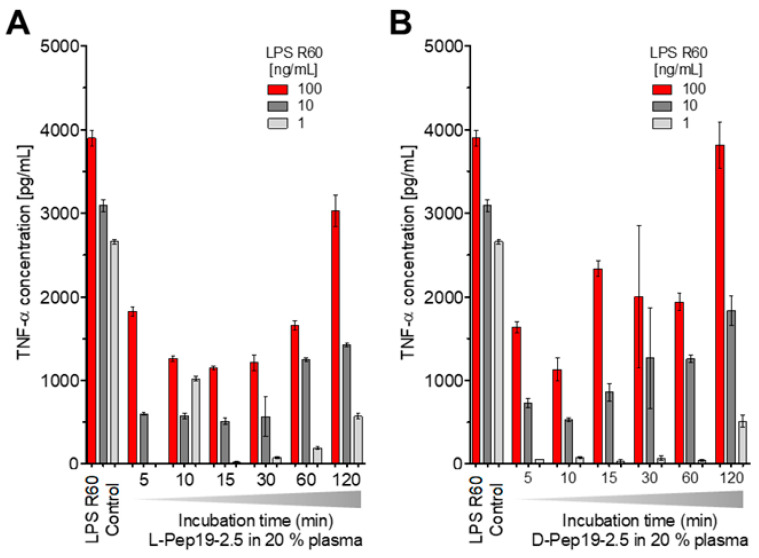
LPS-induced secretion of TNFα by human mononuclear cells in the presence of Pep19-2.5 ((LPS):(Pep19-2.5) 1:10 weight %) incubated before addition at 37 °C for different times in human plasma (20%). (**A**) Data for the L-configured peptide, (**B**) Data for the D-configured peptide which is insensitive to the action of proteases. Mean values and standard deviations due to the determination of TNFα in duplicate.

**Figure 8 ijms-22-01465-f008:**
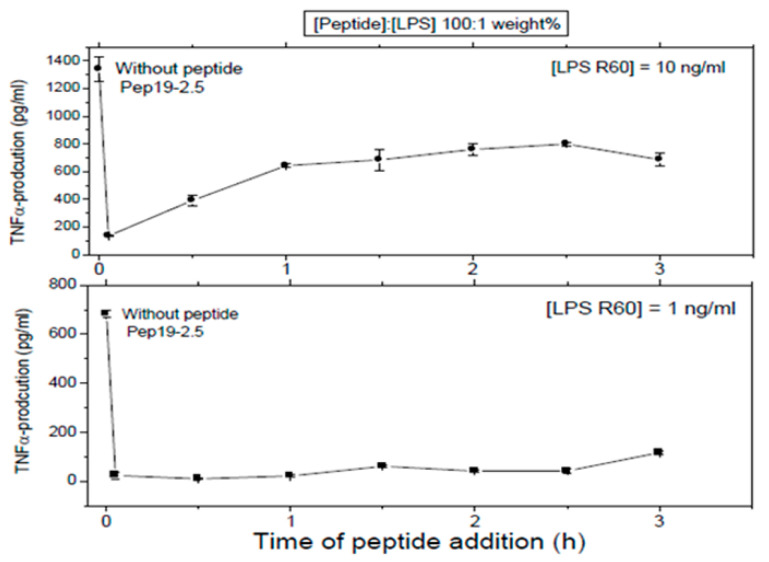
Production of TNFα by MNC cells that were first exposed to LPS and then incubated with peptide Pep19-2.5 [LPS R60]/[Pep19-2.5] 1:100 weight % at different time points from 0 to 3 h after LPS exposure. Mean values and standard deviations due to the determination of TNFα in duplicate.

**Figure 9 ijms-22-01465-f009:**
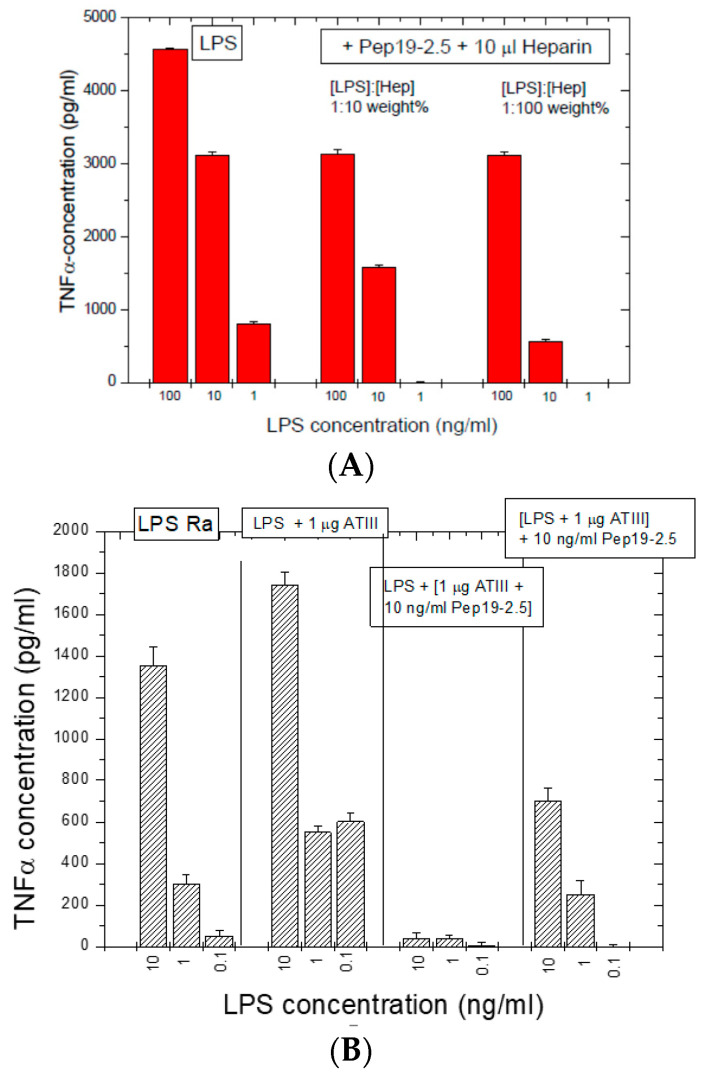
(**A**) Influence of heparin on the inhibition efficiency of Pep19-2.5. The amount of 10 μL corresponds to 50 I.U. (international units). LPS was added at the given concentration to 5.0 × 10^6^ hMNC, directly after the heparin and then, the peptide was added. Stimulation was performed for 4 h and then levels of TNF determination were performed. Mean values and standard deviations due to the determination of TNFα in duplicate. (**B**) Influence of anti-thrombin (AT)-III on the inhibition efficiency of Pep19-2.5 in the LPS-induced stimulation of human mononuclear cells.

**Figure 10 ijms-22-01465-f010:**
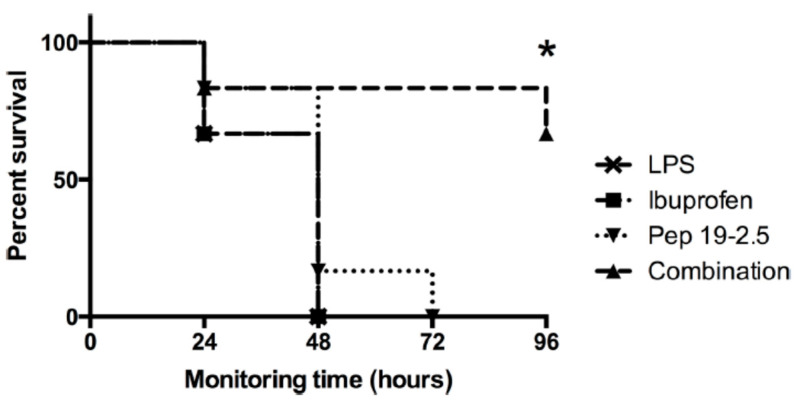
Pep19-2.5 and ibuprofen act synergistically against a lethal dose of LPS in a mouse model of endotoxemia. Redrawn from [13].

**Figure 11 ijms-22-01465-f011:**
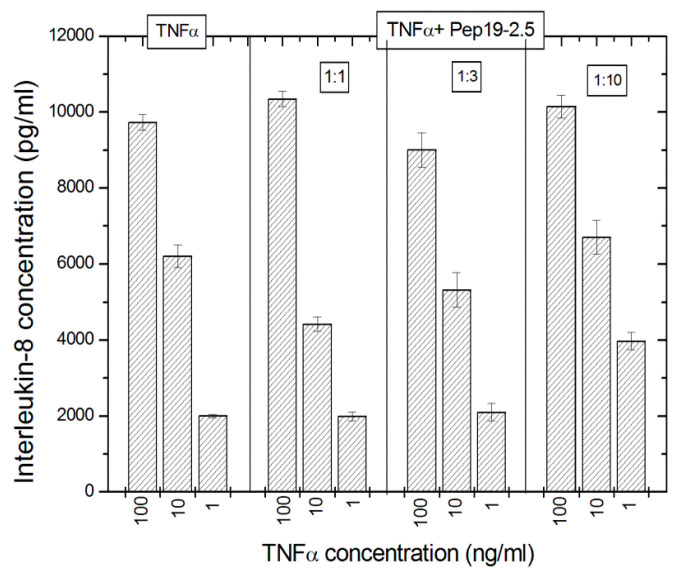
Influence of Pep19-2.5 on tumor necrosis factor α-induced stimulation of human mononuclear cells (10^6^/mL) to secret interleukin-8. Mean values and standard deviations due to the determination of TNFα in duplicate.

**Figure 12 ijms-22-01465-f012:**
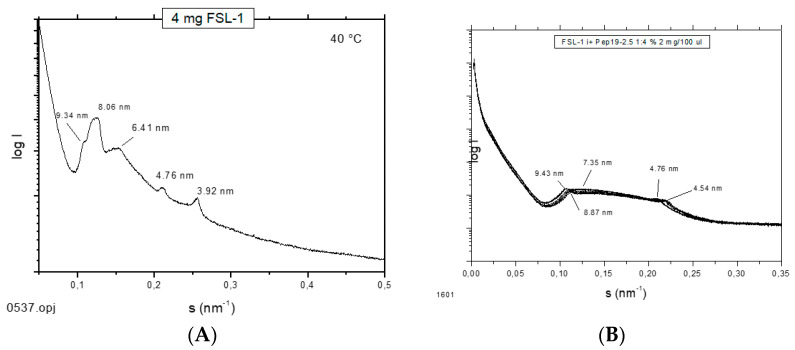
Small-angle X-ray scattering patterns at 40 °C for FSL-1 in the absence (**A**) and presence (**B**) of Pep19-2.5. The logarithm of the scattering intensity log I is plotted versus the scattering vector s (=1/d, d = spacings of the reflections).

**Figure 13 ijms-22-01465-f013:**
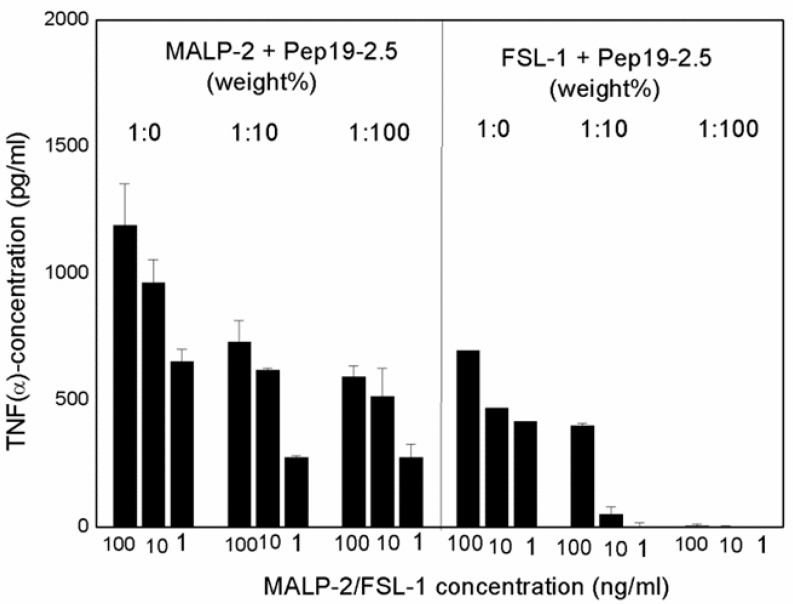
Stimulation of hMNC by macrophage-activating lipopeptide-2 (MALP-2) and fibroblast stimulating lipopeptide-1 (FSL-1) in the presence of increasing concentrations of Pep19-2.5. Mean values and standard deviations due to the determination of TNFα in duplicate.

**Figure 14 ijms-22-01465-f014:**
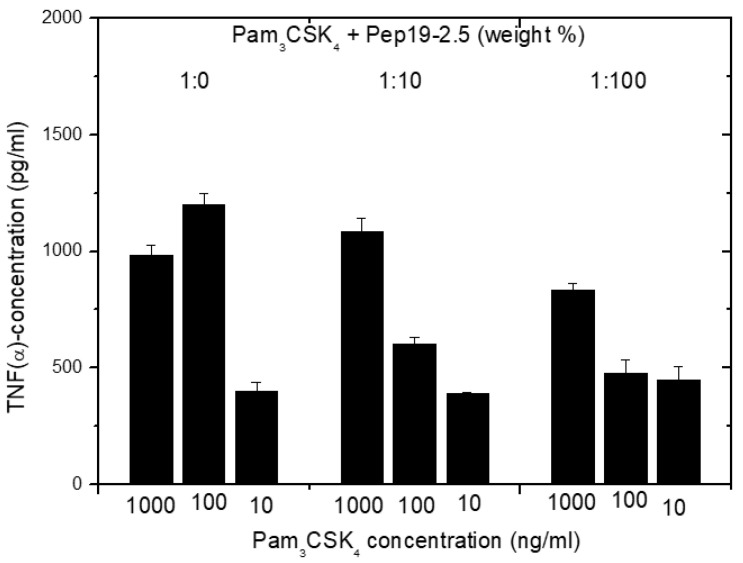
Cytokine induction induced by Pam_3_CSK_4_ in human mononuclear cells at different concentrations of Pep19-2.5. Mean values and standard deviations due to the determination of TNFα in duplicate.

**Figure 15 ijms-22-01465-f015:**
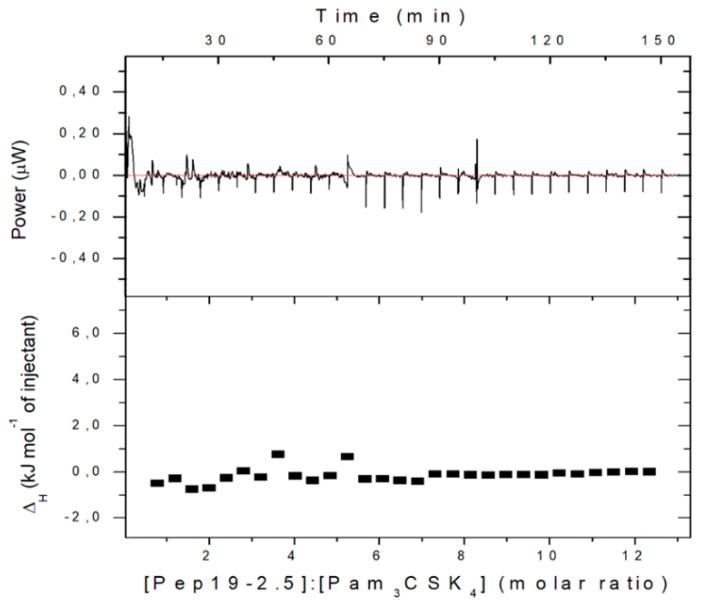
Binding study of Pep19-2.5 with Pam_3_CSK_4i_. The peptide (1 mM) was titrated in 5 μL portions to a 0.05 mM solution of Pam_3_CSK_4_ and the respective enthalpy change was recorded.

**Figure 16 ijms-22-01465-f016:**
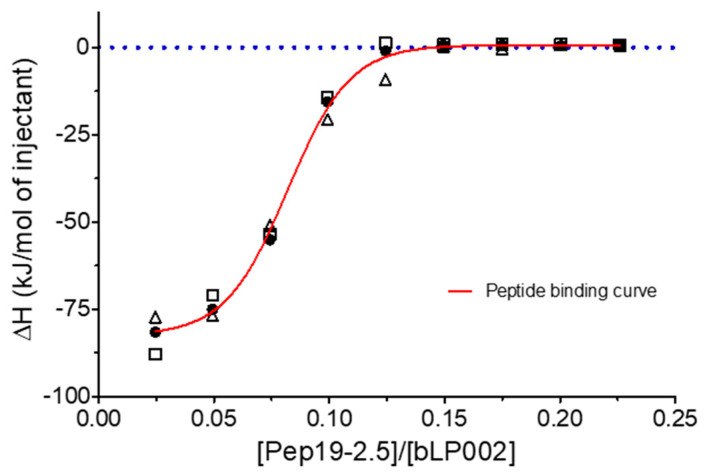
Binding study of the lipopeptide LP with Pep19-2.5 in ITC experiments. The [LP bLP002] = 0.2 mM was titrated with 1 mM Pep19-2.5 using 1 µL in each titration. The data indicate an an S-shaped purely exothermic reaction with saturation.

## Data Availability

The data are available for interested colleagues.

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
