# Peer review of "Anti-Infective and Anti-Inflammatory Mode of Action of Peptide 19-2.5"

_ijms, 2021, doi:10.3390/ijms22031465_

Round 1

Reviewer 1 Report

The manuscript of Heinbockel et al reports the studies on the interaction of polypeptidePep19-2.5 (Aspidasept) with the constituents of the human immune system and with other therapeutically relevant compounds such as antibiotics and non-steroidal anti-inflammatory drugs (NSAID) in order to be able to use this peptide as anti-sepsis drug against the deleterious effects of severe bacterial infections.

The manuscript reports interesting new research on Aspidasept peptide, but some corrections must be made to make it acceptable.

Questions

  • The authors in materials and methods should provide information on the purity of the peptide used.
  • In figure 1, the letters A, B and C shown in the caption are missing.
  • The same problem as in Figure 1 is present in Figure 4. Furthermore, in the first panel 1:5 weight% is not clearly visible.
  • At the line 181 the Authors describe of experiment made to 5 min, 10 min, 15 min, 30 min, and 60 min in 20 % human plasma while in figure 7 the time of 120 minutes is also reported. I ask the authors to correct this incongruity.

Author Response

Thankyou very much. We have adressed all points of the reviewers and have changed the manuscript accordingly. In particular, we have incorporated new Figures 1, 3, and 4. 

Reviewer 2 Report

The manuscript by L. Heinbockel et al., describes the detail analysis of anti-infective and anti-inflammatory mode of Peptide 19-2.5.  Authors politely evaluated the behaviors of Peptide 19-2.5 in the inhibition of lipopolysaccharide-induced cytokine release from mononuclear cells from various viewpoints.  In addition, authors also checked the inhibitory activity of Peptide 19-2.5 against lipopeptide-induced cytokine release.  All experiments including control experiments are well conducted and results are useful for related scientists.  Therefore, I recommend this manuscript as an article in International Journal of Molecular Sciences.  However, some revisions are necessary.

  1. Structural information about lipopolysaccharides used in this study should be shown.
  2. Sequences of Peptide 19-2.5 and 19-5 should be shown.
  3. In immunisation experiments, ITC is not proper to evaluate whether the peptide has immunogenicity. In my understanding, ELISA is a more widely used method to detect antibodies in serum after immunization.  Authors should consider.
  4. The quality of figures are not good. For example, there are no indication of (A), (B), (C) in Figure 1 and Figure 4.  In these figures, it is difficult to understand what three bars mean in each concentration of LPS.  Authors should use the style easy to understand.  In Figure 3, the temperature condition of (C) is 75℃, although figure caption mentioned 40℃.
  5. In several experiments, LPS and peptide were used as weight concentration. However, in Figure 6, LPS and peptide were used as molar concentration.  Authors should same concentration unit.

Author Response

Thankyou, we have changed the Figures 1, 4, and 6. 

Regarding the ITC experiment (point 3 of the reviewer) , we agree with the reviewer. However, the ITC control experiments gives statements about the possible existence of antibodies which then would lead to a high-affinity reaction with considerable enthalpy lacking here.